# Hedgehog Signaling Pathway Orchestrates Human Lung Branching Morphogenesis

**DOI:** 10.3390/ijms23095265

**Published:** 2022-05-09

**Authors:** Randa Belgacemi, Soula Danopoulos, Gail Deutsch, Ian Glass, Valérian Dormoy, Saverio Bellusci, Denise Al Alam

**Affiliations:** 1The Lundquist Institute for Biomedical Innovation at Harbor-UCLA Medical Center, Torrance, CA 90502, USA; randa.belgacemi@lundquist.org (R.B.); soula.danopoulos@lundquist.org (S.D.); 2Department of Laboratory Medicine and Pathology, Seattle Children’s Research Institute, Seattle, WA 98105, USA; gail.deutsch@seattlechildrens.org; 3Department of Pediatrics, University of Washington School of Medicine, Seattle, WA 98105, USA; ianglass@uw.edu; 4University of Reims Champagne-Ardenne, Inserm, P3Cell UMR-S1250, SFR CAP-SANTE, 51092 Reims, France; valerian.dormoy@univ-reims.fr; 5Excellence Cluster Cardio-Pulmonary System (ECCPS), Universities of Giessen and Marburg Lung Center (UGMLC), Justus-Liebig-University Giessen, German Center for Lung Research (DZL), 35392 Giessen, Germany; ed.nesseig-inu.dem.erenni@icsulleb.oirevas

**Keywords:** Hedgehog pathway, development, human lung, branching

## Abstract

The Hedgehog (HH) signaling pathway plays an essential role in mouse lung development. We hypothesize that the HH pathway is necessary for branching during human lung development and is impaired in pulmonary hypoplasia. Single-cell, bulk RNA-sequencing data, and human fetal lung tissues were analyzed to determine the spatiotemporal localization of HH pathway actors. Distal human lung segments were cultured in an air-liquid interface and treated with an SHH inhibitor (5E1) to determine the effect of HH inhibition on human lung branching, epithelial-mesenchymal markers, and associated signaling pathways in vitro. Our results showed an early and regulated expression of HH pathway components during human lung development. Inhibiting HH signaling caused a reduction in branching during development and dysregulated epithelial (SOX2, SOX9) and mesenchymal (ACTA2) progenitor markers. FGF and Wnt pathways were also disrupted upon HH inhibition. Finally, we demonstrated that HH signaling elements were downregulated in lung tissues of patients with a congenital diaphragmatic hernia (CDH). In this study, we show for the first time that HH signaling inhibition alters important genes and proteins required for proper branching of the human developing lung. Understanding the role of the HH pathway on human lung development could lead to the identification of novel therapeutic targets for childhood pulmonary diseases.

## 1. Introduction

Human lung development is divided into five stages, starting at 4 weeks (wks) gestation and continuing through early adulthood when the optimal respiratory function is acquired [1,2,3]. The conducting airways are established mainly during the embryonic (7 wks gestation) and pseudoglandular (7–17 wks gestation) stages, whereas formation of the distal acini, angiogenesis and thinning of the mesenchyme occurs during the canalicular (17–27 wks) and saccular stages (26–36 wks) [1,4,5]. Lung development is a highly regulated and well-orchestrated process. Branching morphogenesis comprises fine spatiotemporal regulation of cellular processes, ultimately generating the complex stereotypically arborized structure characteristic of the lung [6,7]. This process involves continuous crosstalk between the epithelium and underlying mesenchyme, mediated by several important developmental pathways that have proven to be evolutionarily conserved [6,7,8]. This network of pathways coordinates processes essential for proper lung development, including cell migration, proliferation, and differentiation [9].

Several key pathways in lung morphogenesis have been identified, including FGF, WNT, TGFB, and Hedgehog (HH) [10,11,12,13]. The HH pathway is a highly conserved pathway that is not solely involved in lung development but is also necessary for lung homeostasis [14,15,16,17]. The HH pathway comprises three main ligands: Indian Hedgehog (IHH), Desert Hedgehog (DHH), and Sonic Hedgehog (SHH). Whereas IHH and DHH are restricted to bone and gonadal development, respectively, SHH is ubiquitous and plays an essential role during lung branching morphogenesis [14,18,19,20]. Canonical HH signaling is activated when the ligand binds to its receptor Patched-1 (PTCH1), in turn releasing Smoothened (SMO) receptor and ultimately allowing Gli family transcription factors (GLI1, 2 and 3) to translocate into the nucleus and control the transcription of HH pathway target genes such as *CYCLIN D*, *CYCLIN E*, *MYC*, *GLI1*, *PTCH1*, *PTCH2*, or *HHIP*. The HH pathway self-modulates by transcribing negative regulators, such as the co-receptor HHIP (Hedgehog Interacting protein), which competes with the receptor PTCH1 to capture SHH [15,18,19]. This regulation appears important during early lung development since *PTCH1* is expressed in the mesenchyme surrounding the distal tips and at the base of the lung buds, mirroring the *SHH* expression pattern [14]. Likewise, *HHIP* is expressed in the mesenchyme surrounding proximal epithelial regions, where *SHH* is highly expressed [14,20,21,22].

HH signaling is required for embryonic lung development as it regulates branching morphogenesis and mesenchymal proliferation [14,15,23]. *Shh* null mice fail to generate lungs, whereas loss of *Hhip* leads to embryonic lethality due to respiratory distress [14,24]. In mouse lung development, SHH is known to positively regulate epithelial and mesenchymal growth by suppressing FGF10 and upregulating FGF7 [23,25]. Impaired regulation of the HH pathway disrupts epithelial-mesenchymal interactions, ultimately resulting in defective branching that could result in congenital or adult lung diseases. Therefore, several major respiratory diseases, such as lung fibrosis, asthma, and chronic obstructive pulmonary disease (COPD) feature, altered HH pathway activity [17,26,27,28]. However, most of our knowledge of the HH pathway during lung development is inferred from rodent models, with recent studies demonstrating several differences between mouse and human lung structure and function [29,30]. Thus, little is known about the role of the HH pathway in early human lung development. In this study, we used early-stage human lung samples to characterize HH pathway signaling during human lung development. Our data is the first to demonstrate that HH signaling is spatiotemporally regulated during the early stages of human lung development and contributes to epithelial-mesenchymal crosstalk. Dysregulation of the HH pathway disrupts both cell proliferation and programmed cell death and impairs the balance between several signaling pathways (FGF and WNT), resulting in a failure to conduct airway branching.

## 2. Results

### 2.1. Hedgehog Signaling Is Spatially and Temporally Regulated during Human Lung Development

The spatial and temporal expression of HH pathway components in the human developing lung is yet to be determined. As such, we assessed the expression of HH pathway constituents between 10 and 20 wks of gestation (spanning mid-pseudoglandular to mid-canalicular stages of development) using publicly available RNA-sequencing and single-cell RNA sequencing (scRNAseq) data. Bulk RNA sequencing on a native human fetal lung at different gestational stages (10.5 wks = mid-pseudoglandular; 14 wks = late-pseudoglandular; 20 wks = mid-canalicular) was previously published by our group [31]. This dataset showed that the HH pathway components were differentially expressed during human lung morphogenesis. The expression of the main ligand *SHH* increased between 10 wks and 20 wks (respectively 4.54 ± 0.5 and 7.562 ± 0.4 RPKMs; *n* = 3; *p* = 0.0150; Figure 1A) while the transcription factors *GLI1*, *GLI2* and *GLI3* decreased significantly as lung development progressed (Appendix A–C). Moreover, the expression of the receptor *PTCH1* decreased with progression of lung development (2.165 ± 0.1 RPKMs at 20 wks versus 4.14 ± 0.2 RPKMs at 10 wks; *n* = 3; *p* = 0.0025; Figure 1B). In contrast, the expression of *HHIP,* the inhibitor receptor, increased as lung development progressed (15.16 ± 0.9 RPKMs at 20 wks versus 5.42 ±  0.8 RPKMs at 14 wks; *p* = 0.0016 and, versus 6.25 ± 1.1 RPKMs at 10 wks; *p* = 0.0040; *n* = 3 for each group; Figure 1C). Lastly, the expression of the PTCH1 co-receptor *SMO* increased at 14 wks to then decrease at 20 wks (Appendix A).

To better identify cell populations involved in HH pathway signaling, we next evaluated the expression of the primary HH pathway actors in different cell subtypes at different gestational stages, using publicly available scRNAseq data [32]. *SHH* was highly expressed by epithelial cell populations, including basal cells, bud tip and multiciliated cells (Figure 1D). The transcription factors *GLI1* and *GLI2* (Appendix A) were mostly expressed by mesenchymal cells such as airway smooth muscle cells (ASM), while *GLI3*, known as the main inhibitor of HH signaling [33,34], was expressed in both epithelial and mesenchymal cell types with particularly strong expression at 15 wks in R-SPONDIN2 positive (RSPO2+) mesenchymal cells (Appendix A). *SMO* expression distribution was similar to that of *GLI3* (Appendix A). Both *PTCH1* and *HHIP* receptors were strongly expressed by ASM and mesenchymal RSPO2+ cells throughout the developmental stages (Appendix A).

RNA sequencing and scRNAseq results were confirmed by FISH performed on human fetal lung tissues at 11 wks and 18 wks (Figure 1G–I and Appendix A). At both stages, *PTCH1* and *HHIP* were mostly present in the mesenchyme compared to the epithelium. However, *HHIP* expression was strongly restricted to ACTA2 positive cells (mesenchymal progenitors) as described previously [31], while *PTCH1* expression was mainly found in the mesenchymal cells in the vicinity of the epithelial bud tip progenitors. Finally, *SHH* is predominantly expressed in the epithelium. These results, taken together, show differential expression of HH pathway components during early human lung development as well as a specific spatial and cell-type distribution.

### 2.2. Hedgehog Pathway Inhibition Compromises Branching Morphogenesis

To better understand the role of the HH pathway in human lung development, we tested the effect of HH pathway inhibition on lung branching morphogenesis. Human fetal explants within the mid-pseudoglandular stage of gestation (11–13 weeks) were cultured at an air-liquid interface for 48 h as previously described [6,10] and treated with HH pathway inhibitor 5E1 or with negative IgG as control (Appendix A). 5E1 is an antibody that binds the SHH ligand released by the cells and is present in the media [17,35]. Three concentrations were initially tested at 5, 10, and 25 µg/mL. Both 10 and 25 ug/mL concentrations resulted in the death of explants by 48 h (data not shown). Therefore, 5 µg/mL was used for all subsequent experiments. The cultured explants were imaged at 24 and 48 h (Figure 2A), and branching was quantified. 5E1 caused a 50% decrease of branching compared to control (respectively *p* = 0.0007 at 24 h and *p* = 0.0002 at 48 h, *n* = 8 for each group; Figure 2B). In addition, cyst dilatation was observed in the explants treated with 5E1 suggesting a disruption in branch ramification (132.2 ± 4.1% in 5E1 versus 97.6 ± 4.8% in CTL; *n* = 8; *p* = 0.0003; *n* = 8; Figure 2C). Gene and protein expression analyses were performed to confirm HH signaling inhibition. Explants cultured with 5E1 showed a 2-fold decrease in *SHH*, *PTCH1*, and *HHIP* transcript levels compared to control (respectively *p* = 0.0238 for *SHH*, *p* = 0.0152 for *PTCH1*, and *p* = 0.0401 for *HHIP*; Figure 2D–F), while no difference was noted for *SMO* expression (Appendix A). Gene expression of the transcription factor *GLI2*, the main HH signaling, presented no change with 5E1 treatment (Appendix A), while the protein level, evaluated by western blot, was strongly decreased (7-fold compared to CTL; *p* = 0.0105) (Figure 2G,H). These results demonstrate that inhibition of the HH pathway during human lung development triggers branching anomalies.

### 2.3. HH Pathway Regulates Cell Proliferation and Cell Death in Human Lung Development

To identify the processes by which HH signaling impact branching in early human lung development, we first evaluate cell proliferation and cell death since both are required for proper lung branching. We quantified cell proliferation on total tissue as well as epithelial and mesenchymal compartments separately using immunostaining for KI67 and CDH1 (Figure 3A,B). The total number of proliferating cells in the whole tissue showed no significant difference between 5E1-treated explants and control (Figure 3C). However, based on CDH1 staining, epithelial cells displayed significantly more KI67 positive cells in treated explants compared to control (53.80 ± 4.8% in 5E1 versus 26.90 ± 3.5% in CTL; *n* = 8; *p* =  0.0006; Figure 3D). In contrast, the mesenchymal compartment presented significantly less KI67 positive cells in 5E1 treated explants (Figure 3E) compared to control (41.44 ± 3.1% in 5E1 versus 68.48 ± 4.6% in CTL; *n* = 8; *p*  =  0.0003). We next investigated cell death using cleaved-caspase-3 staining (Figure 3F,G). Quantitative analysis demonstrated a global increase of cleaved-caspase-3 positive cells on 5E1-treated explants compared to control (3.44 ± 0.8% in 5E1 versus 0.559 ± 0.1% in CTL; *n* = 4; *p* =  0.0286; Figure 3H). Interestingly, epithelial cell populations did not show significantly altered cell death (Figure 3I) whereas mesenchymal cleaved-caspase-3 positive cells were significantly increased with 5E1 treatment (2.98 ± 0.7% in 5E1 versus 0.403 ±  0.1% in CTL; *n* = 4; *p*  =  0.0286; Figure 3J). These results demonstrated that blocking the HH pathway during early human lung development altered cell turnover by reversing the proliferation ratio between the epithelium and the mesenchyme and inducing mesenchymal cell death.

### 2.4. Hedgehog Signaling Controls the Balance of Progenitor Cells during Human Lung Development

Lung progenitor cells initially play an important role in lung branching morphogenesis by determining proximal-distal patterning. These progenitor cells express SOX2 and SOX9 in the epithelium and ACTA2 in the mesenchyme. We previously demonstrated that a strict spatiotemporal distribution of ACTA2, SOX2, and SOX9 is necessary for human lung branching [6]. Thus, to decipher the mechanisms by which the HH pathway contributes to branching, we examined the progenitor cell makers’ expression and localization on explants treated or not with 5E1. Transcript levels analysis demonstrated an increase of *SOX9* expression in treated explants compared to control (0.001354 ± 0.00014 in 5E1 versus 0.00090 ± 0.00010 in CTL; results expressed in 2-Δct; *n* = 6; *p*  = 0.0390) while no significant changes were observed for *SOX2* (Figure 4A,B). In contrast, *ACTA2* expression significantly decreased with 5E1 treatment compared to control (0.044 ±  0.004 in 5E1 versus 0.063 ± 0.005 in CTL; results expressed in 2-Δct; *n* = 6; *p*  =  0.0332; Figure 4C)**.** Protein expression for these markers was also assessed by western blot analysis. Despite a lack of significant increase in *SOX2* gene expression, the protein level increased significantly in treated explants compared to control (1.896 ±  0.1 in 5E1 versus 0.7261 ±  0.13 in CTL; results expressed in ratios of average pixel intensities; *n* = 3; *p*  = 0.0024; Figure 4D,E). SOX9 protein expression was increased by 2.5-fold in 5E1-treated explants compared to control (1.367 ±  0.06 in 5E1 versus 0.5461 ±  0.17 in CTL; results expressed in ratios of average pixel intensities; *n* = 3; *p*  = 0.0266; Figure 4F,G), while ACTA2 protein expression was decreased by 2-fold in 5E1-treated explants compared to control (*n* = 3; *p*  = 0.0482; Figure 4H,I). Finally, immunostaining of SOX2, SOX9, and ACTA2 (Figure 4J,K) showed an increase of SOX2 (Figure 4L) and SOX9 (Figure 4M) positive epithelial cells following treatment with 5E1 (respectively 24.17 ± 0.74% and 9.47 ± 2%) compared to control (respectively 10.96 ± 2.16% and 4.44 ± 1.2%; *n* = 3; *p* = 0.0286). Our results showed that blocking the HH pathway disrupted the distribution of progenitor cells by modulating SOX2, SOX9, and ACTA2 expression and localization, thus contributing to the defective airway branching.

### 2.5. HH Pathway Regulates FGF and WNT Pathways during Human Lung Development

Branching morphogenesis involves reciprocal interactions between the epithelium and the underlying mesenchyme, which is mediated by several pathways. The HH pathway actively participates to tissue crosstalk by modulating cellular and molecular processes in both mesenchymal and epithelial compartments. Additionally, the HH pathway is part of a complex signaling network including TGF, FGF, or WNT pathways. Since FGF10 plays an important role during branching morphogenesis and SHH is required to control its expression and localization in mice [36], we sought to determine the effect of HH pathway inhibition on FGF10 and its receptor FGFR2 in a human lung development model. RT-qPCR analysis revealed an increase in *FGF10* and *FGFR2* transcripts in 5E1-treated explants compared to control (0.002244 ±  0.0003 in 5E1 versus 0.001251 ±  0.0001 in CTL and 0.1460 ±  0.001 in 5E1 versus 0.07594 ±  0.01 in CTL; results expressed in 2-Δct; *n* = 6 in each group; *p* = 0.0022; Figure 5A,B). However, no change was observed for *FGF7, 9* and *18* (Appendix A), factors that have proven important in murine lung development. In mice, the interaction between SHH and FGF10 is under the control of ETV4 and 5 (34). As expected, *ETV4* and *ETV5* expressions significantly decreased when the HH pathway was blocked (respectively 0.01485 ±  0.004 in 5E1 versus 0.03218 ±  0.005 in CTL and 0.0057 ±  0.001 in 5E1 versus 0.0110 ±  0.001; results expressed in 2-Δct; *n* = 6; *p* = 0.0432 and *p* = 0.0152; Appendix A). FISH for *FGF10* and *SHH* were performed (Figure 5C,D) and RNAscope score demonstrated that 5E1-treated explants expressed significantly more *FGF10* compared to control, confirming RT-qPCR results (0.4942 ± 0.07 in 5E1 versus 0.1213 ± 0.08 in CTL; *n* = 3 in each group; *p* = 0.019; Figure 5E). Our data suggest that inhibiting the HH pathway modulated the expression level of FGF10 during early human lung development, similar to that seen in mouse models.

While the WNT pathway actively participates in the epithelial-mesenchymal crosstalk, very few studies describe a possible interaction between WNT and HH pathways during branching morphogenesis [36,37]. Both receptors PTCH1 and HHIP are strongly expressed in the mesenchymal compartment and co-localize with R-SPO2 in RSPO2+ cells (Figure 1 and Appendix A). R-SPO2 is a secreted protein known to activate the WNT pathway and is required for mouse lung morphogenesis [38]. The expression of these receptors is also decreased following 5E1 treatment (Figure 2E,F). Therefore, we assessed the effect of 5E1 treatment on the expression of *RSPO2* and its receptor *LGR4*. RT-qPCR analysis demonstrated a 3-fold decrease of *RSPO2* expression (0.0619 ±  0.01 in 5E1 versus 0.1822 ± 0.05 in CTL; results expressed in 2-Δct; *n* = 6; *p*  =  0.0415; Figure 5F) and a 2-fold decrease of *LGR4* following 5E1 treatment as compared to control (0.01903 ±  0.005 in 5E1 versus 0.03910 ±  0.005 in CTL; results expressed in 2-Δct; *n* = 6; *p*  =  0.0411; Figure 5G). These results were confirmed by FISH with an evident loss of *RSPO2* in 5E1 treated explants as compared to control (Figure 5H,I). RNAscope scores showed less expression of *RSPO2* when explants were treated with 5E1 (0.1135 ±  0.03 in 5E1 versus 0.3514 ±  0.005 in CTL; *n* = 3; *p*  =  0.05; Figure 5J). These results demonstrate that HH signaling modulates both FGF and WNT pathways during human lung branching morphogenesis.

### 2.6. Hedgehog Signaling Is Downregulated in Congenital Diaphragmatic Hernia

Branching anomalies are features of several human diseases, including congenital diaphragmatic hernia (CDH). CDH is a congenital disease with a prevalence of 1/3000 newborns and displaying 20–40% of mortality due to pulmonary hypoplasia [39]. CDH lungs present as small lungs with fewer airway branches and reduced vascularization, suggesting lung development and branching is arrested during the pseudoglandular and canalicular stages of lung development [40,41]. We sought to determine if the branching defects observed in vitro following HH pathway inhibition correlated with human diseases where branching was impaired. We first investigated HH pathway component expression in CDH lung versus histologically normal lung (listed in Table 3). There was a significant decrease of *SHH* in CDH compared to control lung (0.001291 ±  0.0005 in CDH versus 0.005543 ±  0.0007 in CTL; results expressed in 2-Δct; *n* = 3 and 4 respectively; *p* =  0.0085; Figure 6A). Furthermore, *PTCH1* (Figure 6B) and *HHIP* (Figure 6C) expression were downregulated in CDH (00069 ± 0.003 in CDH versus 0.05776 ±  0.01 in CTL; *p* = 0.0090 and 0.01143 ±  0.009 in CDH versus 0.1391 ± 0.03 in CTL; results expressed in 2-Δct; *p* = 0.0186; *n* = 3 for CDH and 4 for CTL). Additionally, a decreasing trend was observed for *SMO* and the transcription factors *GLI1*, *GLI2*, and *GLI3* (Appendix A). In situ hybridization for *PTCH1* and *HHIP* with *SHH* confirmed the downregulation of HH signaling in CDH lung (Figure 6D,E,G,H). RNAscope score confirmed a strong decrease of *SHH* (0.3215 ±  0.05 in CDH versus 0.6407 ±  0.06 in CTL; *n* = 3 for each group; *p* = 0.0199; Figure 6F), *PTCH1* (0.076725 ± 0.02 in CDH versus 0.3896 ±  0.04 in CTL; *n* = 3 for each group; *p*  =  0.0405; Figure 6I) and *HHIP* (0.09689 ±  0.02 in CDH versus 0.2299 ± 0.01 in CTL; *n* = 3 for each group; *p* = 0.0087; Figure 6J) transcripts in CDH. Our data, taken together, suggest a dysregulation of the HH pathway in CDH lungs.

## 3. Discussion

Branching morphogenesis is an important step in lung organogenesis that allows the establishment of the familiar tree-like branched structure ending in alveolar sacs. The branching process consists of continuous and reciprocal interactions between the epithelium and the mesenchyme, coordinated by numerous signaling pathways [14,15]. It has been clearly demonstrated that HH signaling is crucial for lung branching morphogenesis in murine models [14,15]. However, our knowledge of the role and contribution of the HH pathway during human lung development is limited [14]. A previous study on human fetal lungs solely reported gene expression of *SHH*, *PTCH1*, *SMO* and the *GLIs* throughout early lung development [29]. We have now confirmed these previous observations [29] and expanded into reporting spatiotemporal and cell-type-specific expression of HH components, providing new evidence of HH pathway cellular and molecular regulations throughout the different stages of human lung development (Figure 1 and Appendix A). We also provided the first evidence of the crucial role of the HH signaling pathway in branching morphogenesis using an in vitro experimental system of human lung explant cultures in which we modulated the HH pathway and assessed its role in early human lung development (Figure 2). Here, we unveil a distinct role of the HH pathway as a driver of proper lung development and epithelial-mesenchymal communication by modulating critical processes in human early lung development. Our study focused on the main HH components. However, additional HH regulators such as Suppressor of Fused Homolog (SuFu), which is part of a regulating complex allowing the activation of the HH pathway [42]; Ptch2, Gas1, Cdo, and Boc, all identified as HH linked-receptors but not fully characterized in any physiological context [43,44,45,46], would require further investigations and are beyond the scope of this study.

The HH pathway is often commonly modulated by pharmacologic inhibitors in experimental settings, but their toxicity and off-target effects are often an issue in data interpretation [47,48,49]. There are three main classes of HH pathway inhibitors. The first class targets the SMO pathway receptor. Cyclopamine, a plant-derived teratogen, is among this class. It is clinically used to treat several advanced cancers (brain, lung) [50] but is also widely used to assess the HH signaling [51]. However, recent studies described an apoptosis induction off-target effect of cyclopamine independently of SMO [52,53]. The second class of inhibitors targets the transcription factor GLI1, such as Curcumin or Gant61 [54,55]. The third class contains inhibitors that act on different phases of production, activation, and binding of the SHH ligand, such as RU-SKI 43 (SHH palmitoylation) or 5E1 (binding) [35,56]. Therefore, we used the SHH antibody 5E1, a strong candidate to safely interfere with HH signaling in clinical trials [57] with limited off-target effects, to abrogate HH activation in our human culture model [17,35]. We showed a strong decrease in the number of branches and an evident increase in cyst dilatation upon HH pathway inhibition, suggesting an involvement in lung branching (Figure 2). Evaluating the consequences of 5E1 treatment on the processes orchestrating branching, such as cell proliferation and cell death, suggest that HH signaling inhibition inversely affected cell proliferation in the epithelium and the mesenchyme, while only mesenchymal cell death was affected by HH inhibition. Our data is consistent with studies in mouse explant cultures demonstrating enlarged distal buds following the inhibition of the HH signaling pathway using cyclopamine. These studies also demonstrated an effect of the HH pathway on epithelial and mesenchymal proliferation consistent with our data [58]. Cell proliferation and cell death are known key players in the branching of several organs. We previously demonstrated that a decrease in cell proliferation occurs concomitantly with the transition from a branching program to the canalicular stage of development [6]. Taken together, this suggests that the HH pathway plays an important role in modulating cellular and molecular events necessary for proper lung development in humans.

We previously determined that co-localization of SOX2/SOX9 in the distal epithelial buds and proper distribution of ACTA2+ cells were also required for proper lung branching [6]. Our current results demonstrated an important disruption of SOX2, SOX9, and ACTA2 gene and protein expression following HH pathway inhibition (Figure 4). The epithelial progenitors (SOX2+ and SOX9+) were increased after HH signaling inhibition while ACTA2 expression decreased (Figure 4). It has been previously demonstrated in mice that HH pathway signaling is required for SMC progenitor differentiation during lung development [59]. Thus, HH signaling modulation of progenitor cell expression and localization during human lung development, in addition to the alterations in cell proliferation and cell death, are likely causes of the observed impaired branching.

Whereas the different signaling pathways in lung development and the crosstalk between these pathways have been well-studied in mice [60,61,62,63], little is known about their role in human lung development. To better understand the impact of HH signaling on branching, we evaluated the effects of its inhibition on the other pathways known to play an important role in branching morphogenesis [4,64]. The FGF pathway is one of the most important pathways in mouse lung development, playing a role in the formation of the primary and the secondary bud [9,65]. Several studies in mice show that SHH restricts the expression and localization of FGF10 within the distal mesenchyme to allow for proper branching morphogenesis [37,66]. Upon HH pathway inhibition, we show a significant increase of FGF10 and its receptor FGFR2 (Figure 5), suggesting a similar regulation of SHH on FGF10. Moreover, a previous study in mice demonstrated that excessive FGF10 resulted in dysregulated epithelial proliferation, consistent with our results described in Figure 3 [59]. However, our results did not show any difference in gene expression for other members of the FGF family involved in lung development, such as FGF7, FGF9, or FGF18 (Appendix A). This could be explained by a lack of interaction between these elements and the HH pathway at this developmental stage, or perhaps it may be attributed to species differences between humans and mice [67].

The single-cell data analyzed in Figure 1D–F and Appendix A demonstrate a strong expression of *PTCH1* and *HHIP* in RSPO2+ mesenchymal cells. The inhibition of the HH pathway resulted in a strong downregulation of *RSPO2* and *LGR4*. These findings are consistent with studies in mice, demonstrating that *Rspo2* null mice display severe lung hypoplasia [38]. RSPO2 plays a crucial role in the activation of the WNT/B-catenin pathway, which regulates epithelial-mesenchymal crosstalk during lung development [12,68,69]. It has been described that in mouse lung development, SHH-BMP-WNT signaling is necessary for proper lung specification [70]. Based on our data, the SHH-WNT interactions also seem to be important in the patterning of the human developing lung. Whether SHH directly or indirectly modulates RSPO2 remains to be investigated.

Aberrant HH signaling is associated with several lung diseases such as COPD, asthma, or PF [17,27,71]. Since the HH pathway is a key player during human lung development, it is also involved in developmental abnormalities leading to congenital diseases. Previous studies in mice described a downregulation of SHH. In CDH, the histological abnormalities observed in the lung are accompanied by lower *SHH* expression as compared to normal lung development [72]; therefore, we investigated HH pathway expression in CDH (Figure 6 and Appendix A). Consistent with previous reports, we demonstrated a downregulation of the expression of *SHH* in CDH [57]. Additionally, we show that *PTCH1* and *HHIP* are also decreased in CDH. While other HH pathway elements, such as the GLI transcription factors, showed a decreasing trend, the data for these were not statistically significant and will require additional samples to be able to draw definitive conclusions. Furthermore, our data demonstrate that SHH activating elements are highly expressed in early human lung development (branching period), whereas inhibitory elements are highly expressed in later stages of development. Therefore, time-sensitive inhibition/activation of the HH pathway is essential in devising therapies for different lung diseases in humans. Thus, inhibiting the HH pathway may be a viable therapeutic option for diseases with aberrant activation of the pathway. However, identifying the specific timeline for such interventions will be crucial as we show that activation of the HH pathway is necessary for normal lung development early on. Finally, while there are multiple HH inhibitors available, we chose to use the one with the least off-target effects. The use of inhibitors that interfere with other elements of the HH pathway could provide more insight into detailed molecular regulation of the HH pathway necessary during branching.

Our data highlight the crucial role of the HH pathway during early human lung development, primarily in the context of lung branching morphogenesis. Further studies are necessary to define better the role of the HH pathway in later stages of human development, such as alveolar formation and cell differentiation, as well as its effects on other cell compartments such as the vasculature. In addition, it would be important to investigate whether HH pathway dysregulation is seen in all types of lung hypoplasia (Oligohydramnios, small thoracic cage, etc.).

## 4. Materials and Methods

### 4.1. Human Fetal Lung Explant Cultures

Human fetal lung explants between 10 and 13 wks gestation were cultured in an air-liquid interface as previously described [6,10]. In brief, lung explants were placed atop a porous polycarbonate membrane and cultured in Dulbecco’s Modified Eagle’s Medium: nutrient mix F-12 (D-MEM/F-12) with 1% FBS. Explants were treated with either HH pathway inhibitor 5E1 (5 µg/mL; DSHB, Houston, TX, USA) or negative control mice IgG (5 µg/mL, NB600-597, Bio-Techne, MN, USA) every 24 h and maintained for 48 h. All explants were collected for either RNA, protein extractions, or fixed for paraffin embedding (Appendix A).

### 4.2. Real-Time PCR Analyses

RNA was extracted using the iNtRon Biotechnology, Inc. Easy-Spin™ Total RNA Extraction Kit (Burlington, MA, USA). RNA was reverse transcribed into cDNA using the Tetro cDNA Synthesis Kit (Bioline, Taunton, MA, USA) according to the manufacturer’s instructions. PCR products were amplified using specific TaqMan gene expression assays (listed in Table 1), Applied Biosystems, Foster City, CA, USA), and the TaqMan Universal PCR Master Mix II (Applied Biosystems). PCR products were detected using the StepONE Plus Real Time PCR System (Applied Biosystems). Each sample was run in triplicate.

### 4.3. Immunofluorescence Staining

Human lungs or explants were fixed overnight in 4% PFA at 4 °C, paraffin-embedded, sectioned, and processed for staining as previously described [6,10]. Antibodies are detailed in Table 2.

### 4.4. In Situ Hybridization

Fluorescence in situ hybridization (FISH) was conducted using the Advanced Cell Diagnostics RNAScope Fluorescent Multiplex Assay (Newark, CA, USA) following the manufacturer’s instructions, with minor adjustments as previously described [6,10,30,31]. RNAscope quantification (RNAscope score) was performed on ImageJ. The number of pixels for a dot corresponding to a single mRNA molecule was measured and searched throughout all the area and then reported to the total cell number. For each group, 5 representative images were quantified.

### 4.5. Quantitative Analysis of Proliferation and Cell Death

5E1, IgG control, and untreated control lung explants were immunostained for either KI67 or Cleaved-Caspase 3 and imaged using a 40× objective. CDH1 (E-CADHERIN) co-staining helped distinguish epithelial from mesenchymal nuclei. Ten images were captured and quantified per sample. Cells were counted using a custom macro in Fiji ImageJ software [73]. Nuclei were thresholded using the Bernsen local thresholding algorithm (radius 10), median-filtered (radius 1), and watershed-segmented. Nuclei of 50 pixels and larger were counted.

### 4.6. Immunoblot Analyses

Cultured explants were lysed on ice in RIPA (Radioimmunoprecipitation assay) buffer supplemented with halt protease (Thermofisher) and phosphatase cocktail inhibitors (P5726 and P0044, Sigma). Twenty µg of protein were loaded on an 8% Bis-Tris Plus precast polyacrylamide gel and run in 1X Bolt MES SDS running buffer (Invitrogen) using the Bolt System (Invitrogen Waltham, MA, USA). Transfers were performed using the iBlot 2 System (Invitrogen) and nitrocellulose gel transfer stacks. After the transfer, membranes were blocked in a 50:50 Odyssey Blocking (LI-COR, Lincoln, NE, USA): TBST (TBS and 0.1% Tween) solution at room temperature for at least 1 h and incubated with primary antibodies for 24 h at 4 °C with continuous agitation (Table 2). The following day, membranes were washed with TBST and incubated with fluorescent secondary antibodies diluted in blocking buffer for 1 h at room temperature. Final detection was obtained by enhanced fluorescence with a Chemidoc MP imaging system (Biorad, Hercules, CA, USA). Densitometry was analyzed using ImageLab software (Version 6.1, Biorad).

### 4.7. Statistical Analyses

Due to the limited number of human samples, non-parametric Mann–Whitney and Kruskal–Wallis tests were used as appropriate. A paired Student’s *t*-test was used to detect significant differences between control and treated explants. The results were considered significant if *p* ≤ 0.05.

### 4.8. Study Approval

The human fetal lungs tissues used in this study were collected under IRB approval (USC-HS-13-0399, CHLA-14-2211 and The Lundquist Institute 18CR-32223-01) and provided to the lab by the USC fetal tissue biobank and the University of Washington Birth Defects Research Laboratory (8 lungs: ages 13–15 wks (Table 3), 18 lungs: ages 16+ wks). They are de-identified, and the only information collected was gestational age and known lung pathologies. All samples used in this study were from healthy/normal fetuses with no known defects. Informed consent was provided for each lung collected and used in this study.

## Figures and Tables

**Figure 1 ijms-23-05265-f001:**
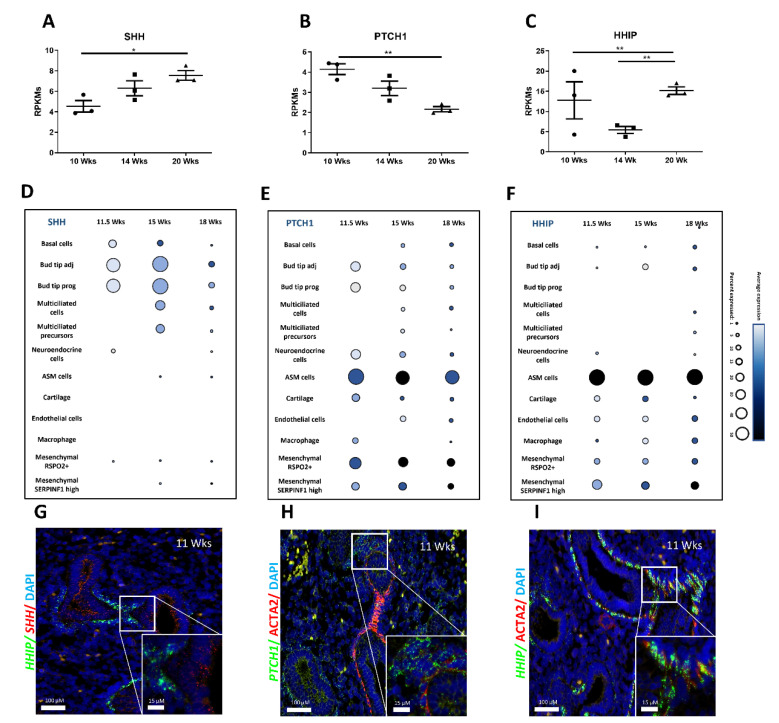
Spatiotemporal expression of Hedgehog pathway components is finely tuned during human lung development. Gene expression of SHH (**A**), PTCH1 (**B**), and HHIP (**C**) expressed in RPKMs ± SEM within the developing lung at 10-, 14- and 20-week gestation (*n* = 3 per time point, * *p* < 0.05, ** *p* < 0.01). (**D**–**F**) Dot plots show the percentage of cells expressing SHH (**D**), PTCH1 (**E**), and HHIP (**F**) using dot size and the average expression level of that gene based on unique molecular identifier (UMI) counts. (**G**–**I**) Fluorescent in situ hybridization (FISH) on 11 wks gestation fetal human lung sections showing SHH (red) and HHIP (green) (**G**); PTCH1 (green) with ACTA2 (IF, red) (**H**) and HHIP (green) with ACTA2 (IF, red) (**I**). Scale bars are 100 µm for large insets and 15 µm for small insets.

**Figure 2 ijms-23-05265-f002:**
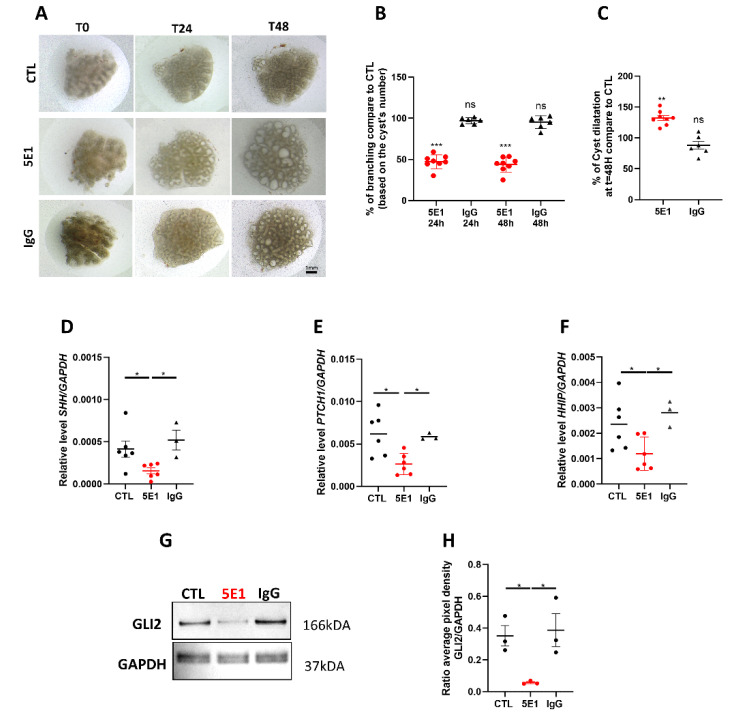
The Hedgehog pathway is required for human lung branching morphogenesis. (**A**) Human fetal lung explant cultures at t = 0, t = 24 h and t = 48 h untreated (in media alone); or treated with 5 µg/mL IgG as negative control or 5 µg/mL 5E1. (**B**,**C**) Graphs showing the percentage of distal branching (**B**) and cyst dilation (**C**) of 5E1, and IgG treated explants as compared to untreated control (CTL) at t = 24 h and t = 48 h. Results are shown in dots and mean ± SEM (*n* = 8 for control and 5E1, *n* = 6 for IgG). (**D**–**F**) RT-qPCR for *SHH* (**D**), *PTCH1* (**E**), and *HHIP* (**F**) performed on control (*n* = 6), 5E1- (*n* = 6; red dots) and IgG- treated explants (*n* = 3). Results are shown as individual data points and mean ± SEM. (**G**) Representative western blots for GLI2 and GAPDH in human fetal lung explants treated or not with 5E1 or IgG (5 µg/mL) for 48 h. (**H**) Dot plots (mean ± SEM) of western blot densitometry ratios for GLI2 normalized to GAPDH (*n* = 3 for each condition). * *p* < 0.05, ** *p* < 0.01; *** *p* < 0.001; ns = no stress.

**Figure 3 ijms-23-05265-f003:**
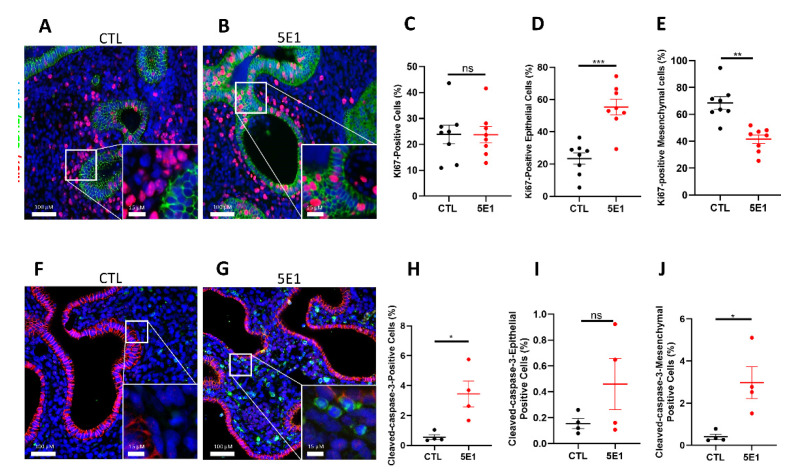
Hedgehog pathway inhibition alters cell proliferation and cell death in human fetal lung explants. (**A**–**D**) IF staining of human fetal lung explants treated with 5E1 for 48 h (**B**) or untreated (**A**) using KI67 (red) and CDH1 (as epithelial marker in green) to assess cell proliferation. Quantification of total cell proliferation (**C**), epithelial proliferation (**D**), and mesenchymal proliferation (**E**). Results are shown as mean ± SEM, *n* = 8 for each group. (**F**,**G**) Cleaved-caspase-3 IF staining of human fetal lungs treated (**G**) or not (**F**) with 5E1 to assess cell death (G vs. F; *n* = 4), Cleaved-caspase-3 is in green; and CDH1 in red. Quantification of total number of Cleaved-caspase-3+ cells (**H**), Cleaved-caspase3+ cells in the epithelium (**I**), and Cleaved-caspase-3+ cells in the mesenchyme (**J**). Results are shown mean ± SEM. * *p* < 0.05; ** *p* < 0.01 and *** *p* < 0.001; ns = no stress.

**Figure 4 ijms-23-05265-f004:**
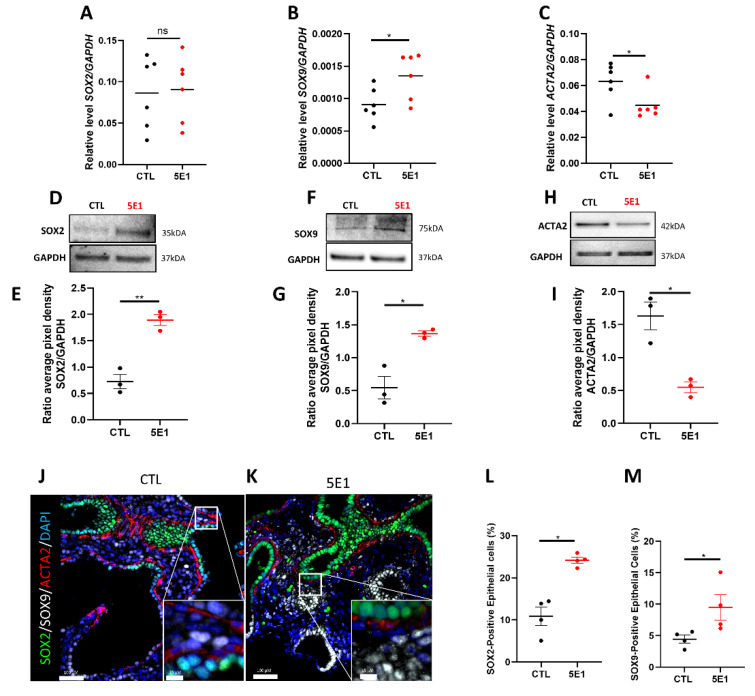
Compartmental cell identity is altered upon Hedgehog pathway inhibition. RT-qPCR for SOX2 (**A**), SOX9 (**B**), and ACTA2 (**C**) in fetal lung explants treated with 5E1 compared to control. Results are shown as mean ± SEM, * *p* < 0.05, *n* = 6 for each group. (**D**–**H**) Western blot analysis of protein expression of SOX2 (**D**,**E**), SOX9 (**F**,**G**), and ACTA2 (**H**,**I**) in 5E1 treated explants compared to control (*n* = 3 for each condition). Western blot densitometry ratios are shown in (**E**,**G**,**I**) (*n* = 3 for each condition). Results are shown as mean ± SEM. IF staining of fetal lung explants treated (**K**) or not (**J**) with 5E1 for SOX2 (green), SOX9 (white), and ACTA2 (red). Quantification of the number of epithelial positive SOX2 cells (**L**) and SOX9 (**M**) (*n* = 4 for each group). * *p* < 0.05, ** *p* < 0.01; ns = no stress.

**Figure 5 ijms-23-05265-f005:**
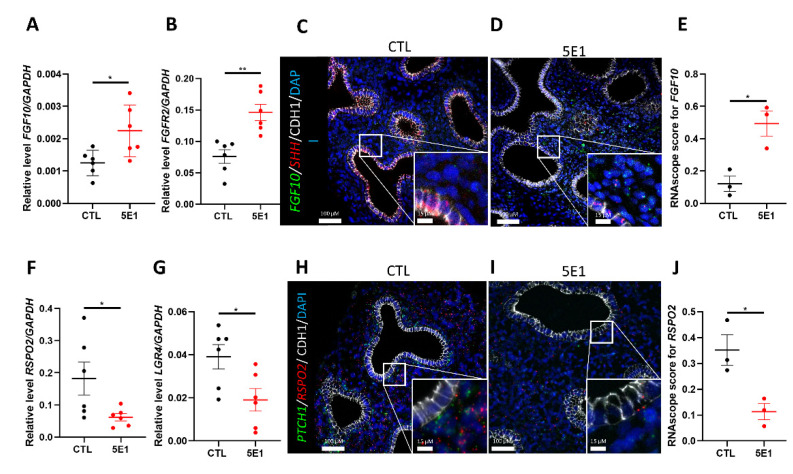
Inhibiting Hedgehog signaling disrupted FGF and WNT pathways. RT-qPCR for *FGF10* (**A**) and *FGFR2* (**B**) in 5E1-treated explants compared to control (results show mean ± SEM, *n* = 6 for each group). In situ hybridization for *FGF10* (green), *SHH* (red), and CDH1 (IF-white) performed on control (**C**) and 5E1-treated (**D**) explants. FISH dots quantification (**E**) revealed more *FGF10* in 5E1 compared to control (results show mean ± SEM, *n* = 3 for each group). RT-qPCR for *RSPO2* (**F**) and its receptor *LGR4* (**G**) on explants treated with 5E1 compared to control (results show mean ± SEM, *n* = 6 for each group). In situ hybridization for *PTCH1* (green), *RSPO2* (red), and CDH1 (IF-white) performed on control (**H**) and 5E1-treated (**I**) explants. (**J**) FISH dots quantification for *RSPO2* confirmed RT-qPCR results (results show mean ± SEM; *n* = 3 in each group). ** p* < 0.05; ** *p* < 0.01.

**Figure 6 ijms-23-05265-f006:**
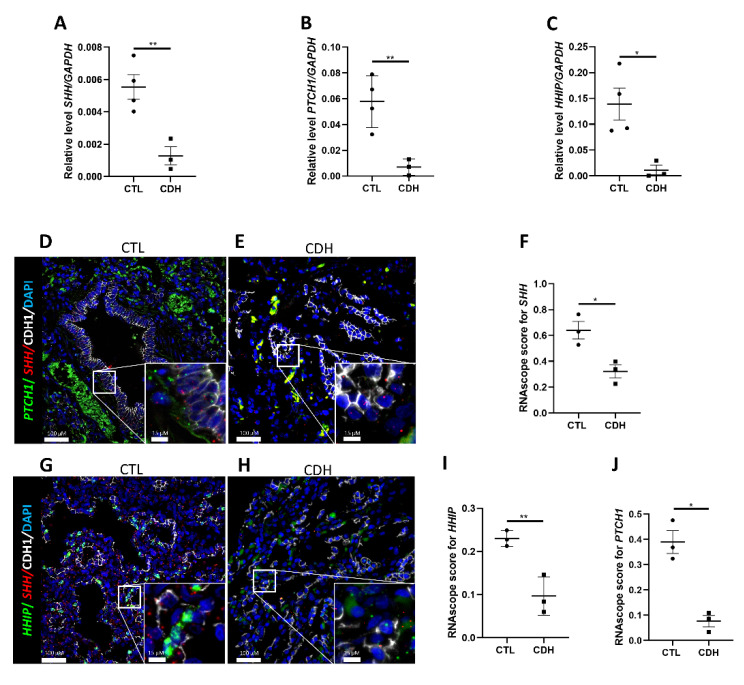
Hedgehog pathway is downregulated in Congenital Diaphragmatic Hernia; (**A**–**C**) RT-qPCR on tissue from patients presenting with Congenital Diaphragmatic Hernia (CDH) and controls for SHH (**A**), PTCH1 (**B**), and HHIP (**C**) (results show mean ± SEM, * *p* < 0.05; ** *p* < 0.01; *n* = 4 for CTL and *n* = 3 for CDH). Combinatorial FISH with IF for either PTCH1 (green, (**D**,**E**)) or HHIP (green, (**G**,**H**)) with SHH (red) and CDH1 (IF-white) performed on tissue sections from CDH (**E**,**H**) and control (**D**,**G**) patients; dot plot quantification (**F**,**I**,**J**) of the FISH signal in CDH vs. control sections (results show mean ± SEM, * *p* < 0.05; ** *p* < 0.01; *n* = 3 for CTL and *n* = 3 for CDH).

**Table 1 ijms-23-05265-t001:** Taqman^TM^ probes used for RT-qPCR.

Target Gene	Probe ID
*ACTA2*	HS00426835
*ETV4*	HS00383361
*ETV5*	HS00927578
*FGF10*	HS00610298
*GAPDH*	HS02786624
*GLI2*	HS01119974
*HHIP*	HS01011015
*LGR4*	HS00173908
*PTCH1*	HS00181117
*RSPO2*	HS04400418
*SHH*	HS00179843
*SMO*	HS01090242
*SOX2*	HS04234838
*SOX9*	HS00165814

**Table 2 ijms-23-05265-t002:** Antibodies used in immunofluorescence and western blot.

Antibody	Company	Catalogue Number	RRID	Host Species	Dilution
ACTA2-Cy3 Conjugated	Sigma-Aldrich St. Louis, MO, USA	C6198	AB_476856	Mouse	1:200
CDH1 (Ecadherin)	BD Biosciences, Franklin Lakes, NJ, USA	610182	AB_397581	Mouse	1:200
Cleaved-Caspase 3	R&D, Minneapolis, MN, USA	AF835	AB_2243952	Rabbit	1:200
GAPDH	Genetex, Irvine, CA, USA	GTX100118	AB_1080976	Rabbit	1:1000
GLI2	Sigma-Aldrich St. Louis, MO, USA	HPA074275	AB_2686677	Rabbit	2 ug/mL (IF) 1 ug/mL (WB)
KI67	Thermofisher, Waltham, MA, USA	RM-9106-S1	AB_2341197	Rabbit	1:200
SOX2	Invitrogen, Waltham, MA, USA	14-9811-80	AB_11219471	Rat	1:100 (IF) 1:500 (WB)
SOX9	Sigma-Aldrich St. Louis, MO, USA	AB5535	AB_2239761	Rabbit	1:100 (IF)
SOX9	R&D, Minneapolis, MN, USA	AF3075	AB_2194160	Goat	1/200 (WB)

**Table 3 ijms-23-05265-t003:** Human samples characteristics for HH pathway characterization in CDH.

Sample Type	Lung Pathology	Age	Gender	Application
CTL	Normal	6 wks	Male	RT-qPCR/FISH
CTL	Normal	28 wks	Female	RT-qPCR
CTL	Normal	1 day	Male	RT-qPCR
CTL	Normal	11 wks	Male	RT-qPCR/FISH
CTL	Normal	23 wks	Female	FISH
CDH	Hypoplasia	23 wks	Female	RT-qPCR/FISH
CDH	Hypoplasia	6 wks	Female	RT-qPCR/FISH
CDH	Hypoplasia	11 wks	Male	RT-qPCR/FISH

## Data Availability

Not applicable.

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
