# Peer review of "Hedgehog Signaling Pathway Orchestrates Human Lung Branching Morphogenesis"

_ijms, 2022, doi:10.3390/ijms23095265_

Round 1

Reviewer 1 Report

This study by Randa Belgacemi et colleagues deals with the role of the HH pathway on human lung development.

Comments and suggestions:

Introduction section: it is difficult to understand the link between lung morphogenesis and HH signaling during embryogenesis. Therefore a summarized scheme on this aspect will be welcomed by the readers.

Also, a diagram with all the steps of this study should be included.

Figure 1: the words are not legible. Revise it.

Subsection 2.6: add a few data about congenital diaphragmatic hernia.

Line 342: mention the  pharmacologic inhibitors used in experimental settings which modulate the HH pathway and their mechanisms

Mention the limitations of this study.

What perspectives for humans does this MS have?

Consider revision accordingly.

Author Response

Reviewer 1

Comment to the Author :

  • Introduction section: it is difficult to understand the link between lung morphogenesis and HH signaling during embryogenesis. Therefore, a summarized scheme on this aspect will be welcomed by the readers.

Response: We appreciate the comment from the reviewer. We kept this section short as SHH pathway is very well known in the mouse lung development. To satisfy the reviewer, we added another sentence as follows:

Line 69-71 “In mouse lung development, SHH is known to positively regulate epithelial and mesenchymal growth by suppressing FGF10 and upregulating FGF7 (25,26)”.

We felt that a schematic on a known pathway might be more appropriate for a review rather than a research manuscript.

  • Also, a diagram with all the steps of this study should be included.

Response: we included a diagram per the reviewer’s request that summarizes the methodology that we included in supplementary figures (Figure 5S).

  • Figure 1: the words are not legible. Revise it.

Response to reviewer:

We thank the reviewer for bringing to our attention this important detail, Figure 1 was ameliorated and replaced in the MS.

  • Subsection 2.6: add a few data about congenital diaphragmatic hernia.

Response: We have now added a few sentences on CDH in section 2.6 as follows:

Line 296-300  CDH is a congenital disease with a prevalence of 1/3000 newborn and displaying 20-40% of mortality due to pulmonary hypoplasia (64). CDH lungs present as small lungs with less airway branches and reduced vascularization, suggesting lung development and branching is arrested during the pseudoglandular and canalicular stages of lung development (65,66)”

Line 342: mention the pharmacologic inhibitors used in experimental settings which modulate the HH pathway and their mechanisms.

Response to reviewer:

Line 352-360: “The first class targets the SMO pathway activator receptor. Cyclopamine, a plant-derived teratogen, is amongst this class. It is clinically used to treat several advanced cancers (brain, lung) [48] but is also widely used to assess the HH signaling [49]. However, recent studies described an apoptosis induction off-target effect of cyclopamine independently of SMO [50,51]. The second class of inhibitors targets the transcription factor GLI1 such as Curcumin or Gant61 [52,53]. The third class contains inhibitors that act on different phases of production, activation, and binding of the SHH ligand such as RU-SKI 43 (SHH palmitoylation) or 5E1 (binding) [36,54].

  • Mention the limitations of this study

Response:

We described three limitations to this study in the discussion :

 - Line 345-349:  However, additional HH regulators such as Suppressor of Fused Homolog (SuFu) which is part of a regulating complex allowing the activation of HH pathway [43]; Ptch2, Gas1, Cdo and Boc, all identified as HH linked-receptors but not fully characterized in any physiological context [44–47], would require further investigations and are beyond the scope of this study.

- Line 423-424: The limitation in the number of control and CDH lung tissues analyzed " showed a decreasing trend, the data for these are not statistically significant and will require additional samples to be able to draw definitive conclusions."

- line 438 : In addition, our data focused on early lung development, it would be interesting to repeat the experimental approaches on late stage of development to complement the observations "to better define the role of the HH pathway in later stages of human development"

According to Reviewer 1 comment, to clarify the MS we added another limitation :

- Line 432-433:  While there are multiple HH inhibitors available, we chose to use the one with the least off target effects. The use of inhibitors that interfere with other elements of the HH pathway could provide more insight into detailed molecular regulation of the HH pathway necessary during branching.

  • What perspectives for humans does this MS have?

Response: CDH lungs in humans present as small lungs with less airway branches and reduced vascularization . These lungs display decreased levels of SHH. In contrast, upregulation of SHH and downstream signaling have been described in bronchopulmonary dysplasia and adult diseases where alveologenesis is compromised such as COPD and pulmonary fibrosis . These findings suggest that SHH signaling is differentially modulated in different human lung diseases, for causes that are yet to be determined. Furthermore, our data demonstrate that SHH activating elements are highly expressed in early human lung development (branching period), whereas inhibitory elements are highly expressed in later stages of development. Therefore, time sensitive inhibition/activation of the HH pathway is essential in devising therapies for different lung diseases in humans. Thus, inhibiting the HH pathway may be a viable therapeutic option for diseases with aberrant activation of the pathway. However, identifying the specific timeline for such interventions will be crucial as we show that activation of HH pathway is necessary for normal lung development early on.

This part was inclued in the manuscript line : 425-432

Reviewer 2 Report

The role of the Hedgehog signaling pathway during prenatal human lung development is to date poorly understood.

This study was conceived to determine the function of the Hedgehog pathway during branching morphogenesis of the human lung.

While confirming previous findings by others, that members of the Hedgehog pathway are expressed during human lung formation, the authors (Belgacemi et al.) of the present study generated novel finding showing the spatiotemporal expression patterns of the Hedgehog signaling machinery.

Using an in vitro organ culture system of human fetal lungs, Belgacemi et al. show that antibody-mediated inhibition of the Hedgehog pathway negatively impacts upon lung branching morphogenesis by altering cell proliferation and cell death in the epithelium and mesenchyme as well as through alteration of the number of epithelial and mesenchymal progenitor cells. The authors also show that the in vitro inhibition of Hedgehog signaling disrupts FGF and Wnt activities in fetal lungs. Furthermore, analyses of lung specimens from human fetuses with congenital diaphragmatic hernia revealed decreased levels of the expression of key players in the Hedgehog pathway. It was concluded that Hedgehog signaling plays a crucial role in driving human lung branching morphogenesis.

Critique

To a large extent, this interesting study is novel. The experiments were well conceived and well performed, and the findings are well presented. Furthermore, the conclusions are based on the high-quality data provided.

Several minor points, however, should be adressed:

  1. Introduction. Line 62: “...which competes with the activator receptor PTCH1...” and line 98: “Moreover, the expression of the activator receptor PTCH1...”. These statements are not correct: PTCH1 is indeed a Hedgehog receptor, however, it is NOT an activator. Rather PTCH1 is an inhibitor of Hedgehog signaling. The Hedgehog signaling cascade is trigerred when a Hedgehog ligand binds PTCH1 receptor. This inhibits PTCH1, leading to SMO release from PTCH1-mediated inhibition.
  2. Results. Line 120: “...was strongly restricted to ACTA2-positive cells...”. Here, the authors should specify that the ACTA2-positive cells are mesenchymal progenitors (although this was done later on in the manuscript).
  3. When describing how lung organ cultures were performed in line 140 in the Results section, the authors refer to reference [6], whereas in Materials and Methods, they refer to reference [10]. This should be clarified/corrected.
  4. Figure 3B: The square showing the magnified area does not include all parts of the tissue shown in the inset.
  5. Legend to Figure 3: “...Cleaved-caspase3+ cells in the mesenchyme (I) and Cleaved-caspase3+ cells in the epithelium (J)” is not correct. It should be “...Cleaved caspase3+ cells in the epithelium (I) and Cleaved-caspase3+ cells in the mesenchyme (J).
  6. Line 268: “These receptors are also decreased following 5E1 treatment (Figure 2F and G)”. This sentence is not adequately constructed. In addition, this sentence contains an error (Figure). It should be corrected to : “The expression of these receptors is also decreased following 5E1 treatment (Figure 2E and F).
  7. Legend to Figure 5, lines 285-286: “RT-qPCR for RSPO2 (G) and its receptor LGR4 (F)...”. This is incorrect and should be “RT-qPCR for RSPO2 (F) and its receptor LGR4 (G)...”.
  8.  Table 2: the text is scrambled in some parts and needs rearrangement.

Author Response

Reviewer 2:

We thank Reviewer 2 for his assessment of our study and we provide here a fully revised manuscript according to his insightful comments. Below are detailed responses to the critiques, point by point.

  • Line 62: “...which competes with the activator receptor PTCH1...” and line 98: “Moreover, the expression of the activator receptor PTCH1...”. These statements are not correct: PTCH1 is indeed a Hedgehog receptor, however, it is NOT an activator. Rather PTCH1 is an inhibitor of Hedgehog signaling. The Hedgehog signaling cascade is trigerred when a Hedgehog ligand binds PTCH1 receptor. This inhibits PTCH1, leading to SMO release from PTCH1-mediated inhibition.

Response: What was meant in these sentences is to state that when Shh binds to PTCH1, it releases SMO and induces the transcriptional activation of the HH pathway. We have now modified these sentences and removed the word activator.

  • Line 120: “...was strongly restricted to ACTA2-positive cells...”. Here, the authors should specify that the ACTA2-positive cells are mesenchymal progenitors (although this was done later on in the manuscript).

Response: We have added, lines 120-121 that ACTA2-positive cells are mesenchymal progenitors.

  • When describing how lung organ cultures were performed in line 140 in the Results section, the authors refer to reference [6], whereas in Materials and Methods, they refer to reference [10]. This should be clarified/corrected.

Response to reviewer: Indeed, these two papers are from our lab and described how lung organ culture is performed. We have corrected this by displaying the two references in line 141 (results) and line 446 (Materials and methods).

  • Figure 3B: The square showing the magnified area does not include all parts of the tissue shown in the inset.

Response: We thank reviewer 2 for his observation, the square in Figure 3B has been corrected and all the others have also been checked.

  • Legend to Figure 3: “...Cleaved-caspase3+ cells in the mesenchyme (I) and Cleaved-caspase3+ cells in the epithelium (J)” is not correct. It should be “...Cleaved caspase3+ cells in the epithelium (I) and Cleaved-caspase3+ cells in the mesenchyme (J).

Response: We thank reviewer 2 for bringing this error to our attention, which has been corrected.

  • Line 268: “These receptors are also decreased following 5E1 treatment (Figure 2F and G)”. This sentence is not adequately constructed. In addition, this sentence contains an error (Figure). It should be corrected to : “The expression of these receptors is also decreased following 5E1 treatment (Figure 2E and F)" Line 272

Response: This has been corrected as requested.

  • Legend to Figure 5, lines 285-286: “RT-qPCR for RSPO2 (G) and its receptor LGR4 (F)...”. This is incorrect and should be “RT-qPCR for RSPO2 (F) and its receptor LGR4 (G)...”.
  • Response: We thank reviewer 2 for bringing this erro to our attention, which has now been corrected.
  • Table 2: the text is scrambled in some parts and needs rearrangement.

Response: We attempted to correct this to the best we can while following the template provided by the journal.

Round 2

Reviewer 1 Report

No answer given.